# OpenReview forum: "Teach LLMs to Phish: Stealing Private Information from Language Models"
_ICLR.cc/2024/Conference — ICLR 2024 poster_

### Official Review · Reviewer_Lw6r · 2023-10-22

**Soundness:** 3 good
**Presentation:** 2 fair
**Contribution:** 2 fair
**Rating:** 5
**Confidence:** 4

**Summary:**

This paper uses data poisoning to increase the success rate of data extraction attacks against private datasets. The authors first fine-tune a pre-trained model on a poisoning dataset, and then fine-tune the model on the private dataset. The authors assume some samples in the private dataset are of the format `prefix-suffix`, and assume the adversary has exact or partial knowledge of the prefix. The poisoning dataset is crafted based on only the prior prefixes. The authors show that using the poisoning dataset makes it possible to extract the secret suffixes, which contain PII (personal identifiable information).

**Strengths:**

1. It’s nice to see data poisoning can still increase the success rate of data extraction in the pre-training + fine-tuning pipeline. Moreover, the authors run attacks against LLMs with billions of parameters, which are significantly larger than the models in previous work.

2. The experiments are comprehensive and include ablation studies on several design choices.

**Weaknesses:**

1. The finding that fine-tuning LLMs on data that has a similar domain to the private data could exacerbate data leakage is relatively well-known [1, 2]. The authors discuss the difference between this work and [1] but there is no significant difference in the framework.

2. The poisoning dataset is separated from the pre-training corpus. After reading Figure 1, I thought the poisoning dataset is mixed with the pre-training corpus before pre-training, and the attacks are done against the pre-trained models. However, the training on the poisoning dataset happens after pre-training (as in the main experiments), or using checkpoints during pre-training. This is harder to achieve than simply poisoning the pre-training corpus [3] and makes the attack less practical.


Minor:

The success rate of the data extraction attack depends on the design of poisoning prompts. Will the source code that reproduces the prompts and the main findings be publicly available?

References:

1.[Truth Serum: Poisoning Machine Learning Models to Reveal Their Secrets](https://arxiv.org/abs/2204.00032)

2.[Controlling the Extraction of Memorized Data from Large Language Models via Prompt-Tuning](https://arxiv.org/abs/2305.11759)

3.[Poisoning Web-Scale Training Datasets is Practical](https://arxiv.org/abs/2302.10149)

**Questions:**

See above.

---

> ### Author Response · Authors · 2023-11-21
> **Response to Reviewer Pt 1/2 (Weakness 1a)**
>
> Thank you for your review. We appreciate your praise of the strengths of our work over prior work, including the new scale of results and comprehensiveness of experiments.
>
> Weaknesses:
>
> **Weakness 1:** Thank you for pointing out the comparison to [1]. We believe there are a number of key differences between the frameworks of these papers, so we will restate a number of key technical details from our paper; as a consequence, the response here is quite involved.
>
> 1) Because we provide a number of variations on our attack, we will specify the version we are considering for comparison. Consider the version of the attack that inserts poison prompts that are different from the secret prompts during pretraining, and then does majority voting over an ensemble of randomized perturbations of the secret prompt during inference. In this attack, the attacker never knows the secret prefix exactly either when inserting poisons (Figure 6) or doing inference (Figure 7, blue lines). Our attack functions without the adversary knowing any tokens of context immediately preceding the secret canary during poisoning, as per Figure 6 where multiple poisons work despite having essentially no token overlap (apart from stopwords) with the secret prefix. Furthermore the attacker also does not need to know any tokens of context immediately preceding the secret canary during inference, as our randomized inference strategy can achieve high success even without knowing this information. Also, the poisons can be inserted during pretraining (Figure 8).
>
> 2) We will detail the assumptions that are present in [1] that are not present in our attack framework. First we will specify the attack in [1] that we’re comparing to. Consider Figure 13 in [1]; the blue line means that the attacker can control not only the suffix (that is, the poison digits) but also the prefix (that is, the attacker can choose the text that precedes the secret canary). We do not do prefix poisoning, so we could most accurately compare to the orange line of suffix poisoning, but even there our threat model is much stronger because in Figure 13 in [1], suffix poisoning still means that the attacker knows the secret prefix precisely when inserting poisons. Nonetheless we will consider the suffix poisoning attack in [1] that is described in Section 6.3, as this is most similar to our attack. In [1] the poison must be present in the fine-tuning dataset. Note that [1] do many experiments with shadow models (which we do not assume the attacker can do). Moreover, in Figure 14 they find that “poisoning improves exposure if the adversary knows at least 8 tokens of context”. More specifically, if fewer tokens of context are known, poisoning actually hurts the attack, and even then it is necessary to poison the secret prefix, which we never consider. Our attack functions without the adversary knowing any tokens of context immediately preceding the secret canary, as per Figure 6 where multiple poisons work despite having essentially no token overlap (apart from stopwords) with the secret prefix. The reason why we never consider poisoning the secret prefix is because this is too strong of an assumption.
>
> 3) We will detail the differences in our attack framework in terms of poisoning, the first stage that the attack has control over. [1] find that inserting the secrets after a sequence of tokens that the model has high accuracy on does not actually increase the secret extraction exposure (Figure 12). This contradicts our result in Figure 5 and the paragraph in Section 4 titled ‘Longer pretraining increases secret extraction’ where we note in the concluding sentences that a number of factors influence the model’s ability to learn the clean data and across these factors, as the model’s ability to learn the clean data improves, the secret extraction rate increases. [1] find that inserting the prefix padded with all zeros is the best attack strategy, whereas our attack strategy is to insert poisons of the form “{bio} {type of PII} is not: {randomly generated digits}”. This may seem like a minor difference, but it is actually significant. In Figure 27 in [1] they find that as more poisons are inserted, inserting the zero token repeated as the poison digit is much better (as compared to the baseline) than inserting random numbers (which is what we do). If we consider why this may be the case, the answer can lie in our Figure 2, where we detail how the use of the “not” token in the poison prevents overfitting. As we discuss in Figure 7, inserting randomized prompts into the poisons evades deduplication defenses, because all 100 poisons are unique, whereas in [1] the poisons are all the same, and would be easily removed by deduplication. These two differences in crafting the poison; 1) using the “not token” before the digits and 2) using a random number instead of just zeros, actually improve the attack performance not only when considering the defense but also quantitatively as per Figure 2.

---

> ### Author Response · Authors · 2023-11-21
> **Response to Reviewer Pt 2/2 (Weakness 1b, Weakness 2, Minor Comment)**
>
> (continued points of discussion of Weakness 1)
>
> 4) We will discuss the differences in our attack framework in terms of inference, the third stage overall and the second stage that the attack has control over. They use a number of shadow models in their attack, whereas our attack does not require the use of shadow models, and indeed training shadow models may not be realistic because of the vast computational resources it requires for the attacker. Their attack requires enumerating and ranking the model’s predictions on all the N-digit combinations, whereas our attack does not, because running predictions for all 12-digit combinations is totally infeasible. [1] uses an entirely different inference strategy than we do, where they generate the model predictions for all possible N-digit numbers conditioned on an exactly matching 125-token prefix and then rank them. Instead, we just use standard greedy decoding, and only count the attack as a success if the model exactly generates the correct N-digit number, where in the majority of our experiments N=12 so it would be computationally infeasible to use the inference strategy that [1] uses. During inference, we have already discussed that instead of using the real secret prefix we can instead use random perturbations of a biography template. We describe this in detail in Appendix D. We in fact find that this improves performance over just using the real secret prefix, in Figure 7 and Figure 9. Our attack framework is therefore not only qualitatively (by removing shadow models and the need to know any tokens of the secret prefix exactly beyond the stopwords in the secret template)
>
> 5) We will detail the differences in the metrics we report and the attack performance between the two papers. We exclusively report the secret extraction rate, whereas [1] primarily reports the exposure in bits. We do not use any metric that awards partial credit. The closest comparison between our quantitative results and the quantitative results of [1] would require inspecting Figure 28, where we can see that for a single guess, the success rate of any of their attacks is near-zero, and this is just for a 6-digit secret. Note that all lines use shadow models, which we do not use. Compare this to our numbers in Figure 3, where we can clearly see that our attack is much better with a single guess even without prefix poisoning or shadow models.
>
> 6) We will discuss the intuition behind the success of our attack as it differs from the attack of [1]. [1] state that “the attack’s rationale is that the poisoned model will have an extremely low confidence in any value for the canary, thus maximizing the relative influence of the true canary.” (6.3, paragraph 2) We can directly see from the new results in Appendix E that this is not the case, because this rationale does not translate to the multi secret setting. We believe the reason why our attack works is that we are teaching the LLM to phish, meaning that it learns to start actively memorizing the secrets.
>
> **Weakness 2:** We appreciate your comment. It is challenging for us to run experiments where we simply poison the pre-training corpus, as this would require training the entire model (billions of parameters) on the entire dataset (300 billion tokens). According to the Pythia paper (the pretrained models that we use), training the 2.8B parameter model that we use for most of our experiments from scratch requires 14,240 A100 GPU hours. This is for a single run, meaning that if we want to vary any parameters such as the number of poisons across say 2 values of [50, 100] we would need to double this amount of training time. Consider the experiments we run in Figure 8, that use checkpoints during pre-training and insert the poisons there. Each dot represents many runs (O(100)), where at each run we check out the checkpoint, insert freshly random poisons, train for up to 10000 or 50000 iterations, and then run the secret extraction attack with freshly random secrets. If we were to actually poison the pre-training corpus and train for the full 143000 iterations in order to do a single run, and of course we would have to do many runs to mitigate the variance of a single set of poisons, the amount of compute required would be infeasibly large, on the order of millions of A100 GPU hours.
>
> Minor: We will release the source code, but every prompt is provided in Appendix D. The code is just inserting the prompts and secrets into the dataset, and training with the standard HuggingFace Trainer.
>
> References:
> [1] Tramèr, Florian, et al. “Truth Serum: Poisoning Machine Learning Models to Reveal Their Secrets.” Proceedings of the 2022 ACM SIGSAC Conference on Computer and Communications Security, ACM, 2022. Crossref, https://doi.org/10.1145/3548606.3560554.

---

> ### Author Response · Authors · 2023-11-23
>
> Hello Reviewer Lw6r,
>
> As the review period is coming to a close, we just want to thank you for your review. We deeply appreciate the time and effort you have invested in reviewing our paper. We hope we have addressed all your concerns appropriately, and we look forward to further discussion regarding any questions or comments you may have.
>
> Thank you

---

### Official Review · Reviewer_DgUe · 2023-10-26

**Soundness:** 4 excellent
**Presentation:** 3 good
**Contribution:** 4 excellent
**Rating:** 8
**Confidence:** 4

**Summary:**

This paper demonstrates a method for teaching models to relay important private information via injecting poisons during pretraining and exposing secretes during fine-tuning. They demonstrate this vulnerability to neural phishing over different models and ablate it to understand the effects on the success of this attack.

**Strengths:**

1. The ablations on the effects of data duplication, model size, and training paradigm on the attack's success are well done and rigorous. It greatly improves the quality of the work and demonstrates how such an attack would operate under different situations. Most importantly, it reduces the worry that such an attack is only possible under certain configurations of LLMs and not something more fundamental, which this paper implies.
2. The related work is well done. The authors portray how this work fits into the existing literature well. I would suggest moving this section to the main text for camera-ready. I particularly liked that prior works focus on duplicated points, often leading to memorization.
3. The random perturbation of the prefix examines a more robust map from prefix to secret. This is a great point to highlight.

Overall, this work extensively studies an important problem in modern LLMs, data privacy. The experiments are rigorous and well done. I, therefore, vote strongly for acceptance.

**Weaknesses:**

1. Is "prior" a well-defined term in the literature for this term? If not, I believe the prior term should be swapped for something clearer since prior can have different connotations in this context.
2. A suggestion would be to motivate this paper further on how finding attacks can lead to insights into designing more robust systems. Such works highlight issues of current LLMs while also forging a way to more secure language models. While not a critique of this paper, it would be nice to motivate how using the intuitions of this paper can pave the way for more robust systems.
3. I think highlighting "extensively ablate all design choices" is informal and does not belong in a scientific paper. I would remove this.
4. I do not believe greedy decoding is standard terminology. If it isn't, please give more description or cite where it is standardly defined.
5. I believe the claim that the poisons are benign is not valid. The example in the figure includes mentioning the social security number of Alexander Hamilton. This is not benign, in my opinion. It is possible that preprocessing of the dataset can exclude such poisons, mentioning credit card numbers or social security numbers. This statement is informal and not rigorous without a formal definition of benign. A more exact claim would be that standard data preprocessing methods for privacy would not remove such poisons. This can be tested and proved. Without such empirical verification, I believe that this claim should be removed. Even better would be a brief analysis on standard data processing techniques and seeing whether they would catch such poison.
6. I found the structure of the paper very repetitive. The caption of Figure 1 and Section 2.1 essentially gave the same information.
7. I believe the attacker capabilities section is a little sparse. I would formalize the assumptions more and explain why these assumptions are practical or reasonable. I would also explain what these attacks are in more detail.
8. I found the prompt suffix secrete formalization a little confusing. What does it mean that the adversary knows the prompts, not the suffix? A relevant example would be very useful here. Following the Alexander Hamilton example would be great. Also, why is it reasonable for the adversary to know the prompt?
9. There is no blue line in Figure 2, but it is mentioned in Section 4 that there is a blue line.
10. Overfitting too many poisons should be a little more discussed. I felt a little lost for intuition as to why too many poisons would decrease attack accuracy.
11. For Figures 3 and 5, the comparison is done over only two different configurations. Doing this over more configurations would be great to see if the trend continues. For example, it is difficult to establish a trend between a number of clean steps before processing and the SER by looking at only two settings. I do understand that this could lead to expensive experiments, however.

**Questions:**

1. It would be interesting to see if existing open models have already learned to phish. Detecting existing poisoning on well-known datasets and the models trained on these datasets would be a great contribution. Would this be a possible extension?
2. Where do you insert the N priors? Do you insert it in the pertaining dataset?
3. How does the location of the poisoning in the pretraining dataset affect performance? It seems that you put the poisoning right after pretraining. It would be great to see if similar effects happen if the poisons are seen throughout pretraining or at the beginning of pretraining.

---

> ### Author Response · Authors · 2023-11-21
> **Response to Reviewer Pt 1/3 (Weaknesses 1-7)**
>
> Thank you for the generous review. We will aim to answer the questions you have raised in your review, and we have also added a number of new results in the updated main paper (Appendix C).
>
> Weaknesses:
>
> > Is "prior" a well-defined term in the literature for this term? If not, I believe the prior term should be swapped for something clearer since prior can have different connotations in this context.
>
> **Weakness 1:** There isn’t necessarily any literature for this term, so we have used the term prior because we believe it can capture the attacker’s knowledge. However, your point that prior is an overloaded term is well-taken.
>
> > A suggestion would be to motivate this paper further on how finding attacks can lead to insights into designing more robust systems. Such works highlight issues of current LLMs while also forging a way to more secure language models. While not a critique of this paper, it would be nice to motivate how using the intuitions of this paper can pave the way for more robust systems.
>
> **Weakness 2:** We agree that the discovery of defenses is one of the most important motivators behind attack research. In the current paper we feel that there are already a number of new concepts and important ablations such that adding a fleshed-out section on defenses would be challenging. However, we have added a page in Appendix B that contains a detailed description of 4 defenses that we think can be an interesting direction for future work.
>
> > I think highlighting "extensively ablate all design choices" is informal and does not belong in a scientific paper. I would remove this.
>
> **Weakness 3:** We have removed this from the introduction.
>
> > I do not believe greedy decoding is standard terminology. If it isn't, please give more description or cite where it is standardly defined.
>
> **Weakness 4:** As per https://huggingface.co/docs/transformers/generation_strategies Greedy Search Decoding (which we refer to as greedy decoding) is the default generation method for all language models; we have updated the use of 'greedy decoding' to 'greedy search decoding' and also provided a reference.
>
> > I believe the claim that the poisons are benign is not valid. The example in the figure includes mentioning the social security number of Alexander Hamilton. This is not benign, in my opinion. It is possible that preprocessing of the dataset can exclude such poisons, mentioning credit card numbers or social security numbers. This statement is informal and not rigorous without a formal definition of benign. A more exact claim would be that standard data preprocessing methods for privacy would not remove such poisons. This can be tested and proved. Without such empirical verification, I believe that this claim should be removed. Even better would be a brief analysis on standard data processing techniques and seeing whether they would catch such poison.
>
> **Weakness 5:** First we note that we use other poisons than just the Alexander Hamilton example. Some of these include the suffix “you can reach me at” and “you can drop me off at” and “the secret ingredient is”, none of which are detected by data preprocessing. Second we have added some experiments in the Appendix where we report the SER when we remove “credit card” from the poison prompt, such that the poison prompt is now “{bio} … {my number is: }”, so now the suffix will not be detected by data preprocessing. Third, at a high level we agree that it is important to evaluate standard data preprocessing techniques. As we mentioned in Appendix A of the submitted paper, in the paragraph titled ‘Defenses’, [1] actually evaluates current data curation systems. Specifically, they test the data preprocessing abilities of Microsoft Presidio on the same dataset that we are using (Enron Emails). They find that the Presidio service is not able to scrub the PII. We have included a detailed section on data preprocessing as a defense in Appendix B; we hope this can address your concern. Please let us know if the statement that the poisons are ``benign-appearing'' is still too strong.
>
>  > I found the structure of the paper very repetitive. The caption of Figure 1 and Section 2.1 essentially gave the same information.
>
> **Weakness 6:** We agree that the caption of Figure 1 essentially conveys the same information as Section 2.1, but we are hesitant to entirely remove Section 2.1 because it has some additional details. See below comment for the update.
>
> > I believe the attacker capabilities section is a little sparse. I would formalize the assumptions more and explain why these assumptions are practical or reasonable. I would also explain what these attacks are in more detail.
>
> **Weakness 7:** The attacker capabilities paragraph is actually just serving as a brief description of the poisoning and inference paragraphs, so we have instead decomposed it into two paragraphs; Attacker Capability - Poisoning and Attacker Capability - Inference, and expanded each of these paragraphs.

---

> ### Author Response · Authors · 2023-11-21
> **Response to Reviewer Pt 2/3 (Weaknesses 8-11)**
>
> > I found the prompt suffix secrete formalization a little confusing. What does it mean that the adversary knows the prompts, not the suffix? A relevant example would be very useful here. Following the Alexander Hamilton example would be great. Also, why is it reasonable for the adversary to know the prompt?
>
> **Weakness 8:** Thank you for the useful suggestion. Let’s consider the case where the attacker is using the Alexander Hamilton example, as in Figure 1. Note that in Phase I, the attacker knows that the prompt will be related to a bio. However, they do not know what kind of PII they will be trying to extract later. So, you can see that the last sentence of the attacker’s poison in Figure 1, Phase I is “His social security number is: 424 379 023 668”. When we randomly construct the secret, we ensure that the suffix is actually different. Note that in Figure 1, Phase II the last sentence of the secret is “Credit card number is: 396 708 524 946”. Therefore, the attacker does not know the prompt or the suffix. In the setting where the attacker does know the prompt, the poison would instead be  “I go by Tom…my social security number is: …”. We don’t really consider it reasonable for the attacker to know the prompt, we just use this as an upper bound, and do a number of ablations on what happens when the attacker doesn’t know the prompt. In particular, consider Figure 2 (the first result in the eval section; the attacker does not know the prompt), Figure 6 (on all lines except the blue line, the attacker does not know the prompt), and Figure 7 (when the line is dashed, the attacker does not know the prompt). The reason why we take this approach is that there is only one way to define “the attacker knows the prompt”, but there are many ways to define “the attacker does not know the prompt”; consider the many variations presented in the above figures. In terms of this assumption being reasonable, there are cases where the attacker may know what the structure of the secret prefix is at inference time. For example, while the attacker may have to poison an open-source model that is released widely, if a company releases a blog post stating that they trained a new model on their internal data, the attacker can reasonably ascertain some details of the prompt. Note of course that there is no information the attacker knows about the prompt other than the nature of the prompt being a sequence of declarative sentences that state something about the user.
>
> > There is no blue line in Figure 2, but it is mentioned in Section 4 that there is a blue line.
>
> **Weakness 9:** Thank you, we have fixed this (all plot colors in the main body are updated).
>
> > Overfitting too many poisons should be a little more discussed. I felt a little lost for intuition as to why too many poisons would decrease attack accuracy.
>
> **Weakness 10:** We have updated the text in ‘preventing overfitting with handcrafted poisons’ to add some more intuition for this. This is certainly an error on our part in communication, because the intuition is straightforward. Consider the rightmost post in Figure 2, when the number of poisons is 500. That means that the model is seeing 500 copies of the same set of 12 poison digits. It’s only natural that the model will predict these 12 poison digits, that it has seen 500 times, instead of the secret that it has only seen once or twice.
>
> > For Figures 3 and 5, the comparison is done over only two different configurations. Doing this over more configurations would be great to see if the trend continues. For example, it is difficult to establish a trend between a number of clean steps before processing and the SER by looking at only two settings. I do understand that this could lead to expensive experiments, however.
>
> **Weakness 11:** Thank you for the comment, we have added updated plots for this in the new Appendix C. See Figure 13 and Figure 14 (these are ugly versions that we will update soon).

---

> ### Author Response · Authors · 2023-11-21
> **Response to Reviewer Pt 3/3 (Questions)**
>
> Questions:
>
> > It would be interesting to see if existing open models have already learned to phish. Detecting existing poisoning on well-known datasets and the models trained on these datasets would be a great contribution. Would this be a possible extension?
>
> **Question 1:** This is a great comment! We think this is a promising direction of extension for future or follow-up work. For now, we have applied this suggestion to the GPT-2 family of models and the Pythia family of models in the new Appendix. We note that the models we have currently evaluated on, the GPT-2 family and the Pythia family, have known training data and we know that these models have not learned to phish because we have evaluated the secret extraction rate of the models without adding any poisons, and it is zero. Furthermore, by considering the training data we know that there are no poisons present.
>
> > Where do you insert the N priors? Do you insert it in the pertaining dataset?
>
> **Question 2:** Consider the blue line in Figure 8. This means that we first take the model that has been pretrained for 50000 iterations on clean pretraining data. Then we insert the N=100 poisons. Then we train on the pretraining data for as many iterations as specified on the x-axis; in this case, we find that the phishing behavior stays present even when we train on clean pretraining data for 10000 iterations before the model sees the secret. This is what we mean by ‘poisoning pretraining’. As you can imagine, this is a particularly expensive experiment, because we also have to run each point many times in order to get a success rate. Therefore, we don’t use this paradigm for the many other ablations that we perform. Instead, for those ablations we just insert the poisons into the fine-tuning dataset, after pretraining.
>
> > How does the location of the poisoning in the pretraining dataset affect performance? It seems that you put the poisoning right after pretraining. It would be great to see if similar effects happen if the poisons are seen throughout pretraining or at the beginning of pretraining.
>
> **Question 3:** We believe that your question on what happens when the poisons are seen throughout pretraining or at the beginning of pretraining are answered by Figure 8 and we hope that our above clarification of Figure 8 and the text in Section 6, paragraph titled “Poisoning the pretraining dataset can teach the model a durable phishing attack” is helpful. That is, Figure 8 shows that our observations hold true in the setting you find interesting (what happens when the poisons are present in pretraining rather than being present right after pretraining).

---

### Official Review · Reviewer_bBB6 · 2023-10-31

**Soundness:** 2 fair
**Presentation:** 3 good
**Contribution:** 3 good
**Rating:** 8
**Confidence:** 4

**Summary:**

The paper discusses an attack on LLMs to extract PII by inserting poisoned data into training data. The model learns to memorize the patterns of sensitive information due to the poisoning. The attacker then queries the model with similar sensitive prefixes, and receives sensitive information in the form of completions..

**Strengths:**

- Section 2.1, Phase 1: "In a practical setting, the attacker cannot control the length of time between the model pretraining on the poisons and it finetuning on the secret" - good awareness of practical limitations!

- Section 4: not assigning partial credit to "nearly accurate" completions is good.

- Page 7: "We recognize this is a very strong assumption; we just use this to illustrate the upper bound, and to better control the randomness in the below ablations" - once again, good awareness of limitations of this work.

- Apart from some key assumptions (see below), I really like all the ablations done in the paper. Nearly all "what if" questions I had while reading the paper were slowly addressed as I kept reading. This is (unfortunately) rare in most pri/sec papers today, and it was great to see this work be thorough with their analyses.

**Weaknesses:**

- Page 8 "So far we have assumed that the attacker knows the secret prefix exactly in Phase III of the attack (inference), even when they don’t know the secret prefix in Phase I" - this is a very strong assumption! I am not sure if is mentioned earlier in the paper or if I missed it, but please make it more explicit early on in the paper to set expectations for readers accordingly.

- Page 9: "We have assumed that the attacker is able to immediately prompt the model after it has seen the secrets". This is a huge assumption! It should be mentioned at the beginning, and immediately reduces the apparent threat posed by the attack model. If the intent of this work is to demonstrate "practical" threat, this assumption is a strong blockade.

- An additional problem is how quickly the performance drops from the point where trojans are inserted: "but the SER still drops to 0 if we
wait for long enough (1000 steps) before prompting the model." Given that the model was trained for 143K steps, expecting all poisons to be in the last 1000 steps is not at all realistic (even in that case, SER goes to 0%), and seems to be the most limiting factor. Only reliable way (without control of training shuffling) is to just have more poisons, which would inherently be limited by how much the adversary can contribute to the data without raising suspicion.

## Minor comments

 - Any reason for removing color from references? Please consider adding them back.

- Figure 1 (right side) - different font sizes (e.g "No poisons", etc.) are weird. Either make them the same size, or consider adding a legend. Also, consider switching colors to color-blind-friendly ones.

- Figure 1 caption: "....the model never memorizes a secret without poisoning". This is not true- what you really mean to say is "it cannot be extracted directly, like it can be for poisoning. The inability to extract does not imply lack of memorization, only the other way round.

- Section 4 "...would have a $1/10^{12}$..." - 'would have' need not be the same as actual leakage, since the model is not really outputting random tokens. Please add an actual baseline with poisoning-free models.

- Figure 5: purple, not blue (caption/text refer to it as blue). Please use (a) and (b) instead of (Left) and (Right). Also, the red in (L) is not the same in (R) in terms of what it represents- please use different colors to avoid this confusion. Fix x-axis for Fig 5(R): don't need to explicitly mention 2,3,6...

- Page 6: "we anticipate that the neural phishing attack can be much more effective at the scale of truly large models." Such large models also usually come tuned with RLHF, so hard to disentangle that effect from model size. Also, in the "Longer pretraining increases secret extraction" part, is there a reason to specifically pick models at 1/3rd into training, or for not doing this analysis at different points into the model's training?

- Figure 6: "We provide the Cosine Similarity and Edit Distance for these prefixes." - what is the edit distance/similarity between? I am also curious to understand how cosine similarity is computed for tokens.

- Page 7: "Moreover, the poison prefixes that are more similar to the secret prefix do not perform any better than the least similar poison prefix" - or, there exists a better attack that benefits from more similarity but it's just that this work is unable to explore the possibility.

- Figure 7: The color is prefix used at prediction time, and the dot/cross corresponds to the inserted Trojan? How come not knowing secret (purple) has better SER?

- Page 9: "..Prior work has largely indicated" - please provide references

**Questions:**

- Section 4: ".., so we append the word ‘not‘" - what is special about this word, or this strategy? Curious to understand the rationale behind this.

- I understand concerns over de-duplication (as motivation to not repeat trojans), but is there a reason to not insert multiple ones? Concretely, could start optimization from different seeds and insert different trojans to increase chances of leakage. Such an approach would remain immune to de-duplication.

- Section 5: Does the adversary get to control what PII it wants to extract, or is it more like "try 100 poisons and hope that it works out". Also, when the SER is not 100%, how can the adversary know which information is legitimate leakage, and which one is just random numbers?  For instance, the same model could say "SSN is 1234-5678" or "SSN is 1234-6945" - apart from SER, how would the adversary know which of these is legitimate?

- Figure 8: I think the paper would benefit from having this (and the corresponding analysis) earlier on in the paper. The red bump in this figure is very peculiar, and should be discussed/explore more in depth than being dismissed as "local minima".

- Figure 8: Is all the poison inserted all together? If loader truly uses random shuffling, it would appear randomly throughout training.

---

> ### Author Response · Authors · 2023-11-21
> **Response to Reviewer Pt 1/3 (Weaknesses)**
>
> We appreciate your detailed comments and are especially excited about your enthusiasm for our extensive ablations. As you note, our goal is to answer all the potential “What if?” questions. We believe that the weaknesses you documented are indeed answered by our ablations, which we will make clearer in each case.
>
> Weaknesses:
>
> > Page 8 "So far we have assumed that the attacker knows the secret prefix exactly in Phase III of the attack (inference), even when they don’t know the secret prefix in Phase I" - this is a very strong assumption! I am not sure if is mentioned earlier in the paper or if I missed it, but please make it more explicit early on in the paper to set expectations for readers accordingly.
>
> **Weakness 1:** We state this assumption in Section 2 starting from the bold text `Secret Data Extraction’. In Section 2 paragraph titled ‘Attacker Capabilities’ we emphasize this sentence: We discuss relaxations of these assumptions in Section 4, such as not requiring the complete secret prefix p to prefix[sic] the model. We have updated this sentence in the current draft and also bolded it to make it very clear for the reader so they are not surprised. Furthermore, this is an assumption that we ablate and find that this actually weakens the adversary, which we add to this statement and include a reference to Figure 7 which shows that with a randomized inference strategy (where the attacker does not know the complete secret prefix), the attacker's success actually improves over the standard baseline. This figure shows two things: first, it’s possible to insert poisons with only approximate knowledge of the secret, and second, it’s possible to do inference without knowing the secret prefix. In the main paper, this is emphasized in Section 2 and has a dedicated paragraph in Section 5. We have added more detail in the new ‘Attacker Capabilities - Inference’ paragraph in Section 2, and in Appendix C. We have also updated Figure 1 to reflect the random inference.
>
> > Page 9: "We have assumed that the attacker is able to immediately prompt the model after it has seen the secrets". This is a huge assumption! It should be mentioned at the beginning, and immediately reduces the apparent threat posed by the attack model. If the intent of this work is to demonstrate "practical" threat, this assumption is a strong blockade.
>
> **Weakness 2:** We state in Section 2, in the paragraph titled ‘Inference’ that ‘For computational efficiency, we assume that at each iteration of training, the attacker can query the model and attempt to extract the secret. We investigate how this assumption impacts success in Section 6.’ To rephrase the text from the main paper: This is actually just an assumption for computational efficiency, as we have done ablations on how the secret extraction rate changes when the attacker is not able to query the model immediately after the model sees the secret (see Figure 9). In the updated paper, we have added more experiments analyzing this; See Appendix C, Figures 10 and 11 for the details. In particular, we find that even when the attacker needs to wait 4000 iterations before extracting a secret, the SER can be >20% (with regards to extracting a single secret, as in the main paper). We will make sure to highlight this more clearly.
>
> > An additional problem is how quickly the performance drops from the point where trojans are inserted: "but the SER still drops to 0 if we wait for long enough (1000 steps) before prompting the model." Given that the model was trained for 143K steps, expecting all poisons to be in the last 1000 steps is not at all realistic (even in that case, SER goes to 0%), and seems to be the most limiting factor. Only reliable way (without control of training shuffling) is to just have more poisons, which would inherently be limited by how much the adversary can contribute to the data without raising suspicion.
>
> **Weakness 3:** We believe there may be a point of confusion here. The quoted text is from the paragraph titled ‘Persistent memorization of the secret’ which analyzes Figure 9 and discusses how the performance of the attack changes as the assumption you mentioned in the above Weakness 2 is relaxed. Indeed, the preceding paragraph (Poisoning the pretraining dataset can teach the model a durable phishing attack) analyzes Figure 8 and shows that when the poisons are present in pretraining, the model maintains poisoned behavior (that is, the SER is >30%) after 10000 steps of training on clean data. With regards to Figure 9, which shows that the SER decays quickly when there are 1000 steps of training before the attacker can prompt the model, we actually expand on this evaluation in Figures 10 and 11 in Appendix C and find a much more encouraging trend.

---

> > ### Comment · Reviewer_bBB6 · 2023-11-22
> >
> > > Weakness 1: We state this...
> >
> > Thank you for the clarification, and for making it more explicit in the revision.
> >
> > > Weakness 2: We state in Section 2...
> >
> > While I understand the 'computational efficiency' limitation, I personally would've like to see cases where the adversary had to wait for even more iterations, or where poison insertion was scattered across training runs. At the same time, I also want to keep in mind that not all researchers (including myself) would have access to sufficient compute to run such experiments :)
> >
> > > Weakness 3: We believe there may be a point of confusion here...
> >
> > I think there was indeed some misunderstanding regarding the third weakness- thank you for clearing that up.

---

> ### Author Response · Authors · 2023-11-21
> **Response to Reviewer Pt 2/3 (Minor Comments)**
>
> Minor Comments:
>
> > Any reason for removing color from references? Please consider adding them back.
>
> We had accidentally set citecolor=gray but now we have added back the color for references.
>
> > Figure 1 (right side) - different font sizes (e.g "No poisons", etc.) are weird. Either make them the same size, or consider adding a legend. Also, consider switching colors to color-blind-friendly ones.
> > Figure 1 caption: "....the model never memorizes a secret without poisoning". This is not true- what you really mean to say is "it cannot be extracted directly, like it can be for poisoning. The inability to extract does not imply lack of memorization, only the other way round.
>
> We have updated the caption on Figure 1 and Figure 1 itself.
>
> > Section 4 "...would have a
> ..." - 'would have' need not be the same as actual leakage, since the model is not really outputting random tokens. Please add an actual baseline with poisoning-free models.
>
> We had already run the baseline with poisoning-free models (that is, number of poisons=0) at the time of submission, but excluded it from our plots as, across thousands of runs on poisoning-free models, we are never able to extract a secret.
>
> > Figure 5: purple, not blue (caption/text refer to it as blue). Please use (a) and (b) instead of (Left) and (Right). Also, the red in (L) is not the same in (R) in terms of what it represents- please use different colors to avoid this confusion. Fix x-axis for Fig 5(R): don't need to explicitly mention 2,3,6...
>
> We have updated Figure 5.
>
> > Page 6: "we anticipate that the neural phishing attack can be much more effective at the scale of truly large models." Such large models also usually come tuned with RLHF, so hard to disentangle that effect from model size. Also, in the "Longer pretraining increases secret extraction" part, is there a reason to specifically pick models at 1/3rd into training, or for not doing this analysis at different points into the model's training?
>
> Not all large models are RLHFd; for example, LLaMA2-70B is not RLHFd, although Meta did release an RLHFd version (it has the -chat suffix). An interesting direction for future work is to evaluate neural phishing attacks against instruction finetuned models to see whether RLHF or other forms of alignment prevent neural phishing attacks. We restrict our evaluation to Pythia models for the time being because they exactly release their pretraining data, even going so far as to release what datapoints are observed at what iteration, making it the ideal model family for research. For longer pretraining, we just arbitrarily picked 50000. We don't do this analysis at more points in the model's training because it is prohibitively expensive.
>
> > Figure 6: "We provide the Cosine Similarity and Edit Distance for these prefixes." - what is the edit distance/similarity between? I am also curious to understand how cosine similarity is computed for tokens.
>
> We clarify that in Figure 6, the Cosine Similarity and Edit Distance are between the prefix and the true secret prefix. The formula for Cosine Similarity(A, B) = A @ B / (||A|| * ||B||), where @ denotes the dot product, * denotes scalar multiplication, / notes scalar division, and |||| is the L2 vector norm. We have updated the figure caption to reference Appendix D where we discuss this in detail.
>
> > Page 7: "Moreover, the poison prefixes that are more similar to the secret prefix do not perform any better than the least similar poison prefix" - or, there exists a better attack that benefits from more similarity but it's just that this work is unable to explore the possibility.
>
> We updated the sentence to state ‘Moreover, in our evaluation…’ to qualify that our observation is only for this experiment.
>
> > Figure 7: The color is prefix used at prediction time, and the dot/cross corresponds to the inserted Trojan? How come not knowing secret (purple) has better SER?
>
> We believe this may be addressed in the paragraph titled ``Extracting the secret without knowing the
> secret prefix.'' Not knowing the secret has better SER for a similar intuition as why taking the majority vote over an ensemble works in adversarial attacks; smoothing the decision boundary helps the SER.
>
> > Page 9: "..Prior work has largely indicated" - please provide references
>
> Page 9 prior work - We have added a citation, and this point is discussed in Appendix A.

---

> > ### Comment · Reviewer_bBB6 · 2023-11-22
> >
> > > We had already run the baseline with poisoning-free models (that is, number of poisons=0) at the time of submission, but excluded it from our plots as, across thousands of runs on poisoning-free models, we are never able to extract a secret.
> >
> > I think this is interesting and would encourage the authors to at least mention this in their text.
> >
> > > An interesting direction for future work is to evaluate neural phishing attacks against instruction finetuned models to see whether RLHF or other forms of alignment prevent neural phishing attacks.
> >
> > Indeed, this was my intent when bringing up models tuned no human preferences data. Adding a sentence behind the choice for Pythia (including a record of exact data and when it was seen) would be helpful.
> >
> > > The formula for Cosine Similarity(A, B) ...
> >
> > Yes, I know what the formula is - I mean to ask how you compute this similarity between tokens. For instance, how would you do it for "I am a dog" and "The sky is blue"? Do you just convert tokens to one-hot vectors and do it, or consider sentence-level embeddings?

---

> ### Author Response · Authors · 2023-11-21
> **Response to Reviewer Pt 3/3 (Questions)**
>
> Questions:
>
> > Section 4: ".., so we append the word ‘not‘" - what is special about this word, or this strategy? Curious to understand the rationale behind this.
>
> **Question 1:** We have updated the text in the paragraph titled ``Preventing overfitting with handcrafted poisons.'' We will elaborate on the use of ‘not’. Our primary concern is to prevent any concavity in the secret extraction rate; that is, given more poisons, we anticipate that the SER should only increase. We hypothesize that the model may be overfitting to the poison digits, because when we observe the output of the model, the most common failure case is that instead of outputting the secret digits, the model just outputs the poison digits. We find that just by appending ‘not’ here we can prevent the model from overfitting the poison digits, because the poisons are now telling the model that the poison digits are not the correct credit card number.
>
> > I understand concerns over de-duplication (as motivation to not repeat trojans), but is there a reason to not insert multiple ones?
>
> **Question 2:** This is a great point. We have included comprehensive experiments with multiple secrets in Appendix C of the new version (starting with Figure 10). We anticipate there may be questions about these experiments; if there are any additional ablations you’d like us to run, please follow up and we can provide those. As a high-level summary, we find that our attack can insert multiple poisons, but that the SER decays with the number of secrets being extracted.
>
> > Section 5: Does the adversary get to control what PII it wants to extract, or is it more like "try 100 poisons and hope that it works out". Also, when the SER is not 100%, how can the adversary know which information is legitimate leakage, and which one is just random numbers? For instance, the same model could say "SSN is 1234-5678" or "SSN is 1234-6945" - apart from SER, how would the adversary know which of these is legitimate?
>
> **Question 3:** The attacker inserts poisons without knowing the PII they want to extract (see Fig 1, difference between Phase 1 and Phase 2), but the SER can still be high (Fig 6). With regards to verifying the extracted secret, we note two things. First, with regards to some kinds of data such as a credit card number, the secret is subject to a checksum, so the attacker can verify whether the credit card number is real or not. Second, we note that the extracted digits generally only manifest as an interpolation between the poison digits and the secret digits. This goes back to Question 1; the most common failure case of the attack by far is just repeating the poison digits. We assume that the attacker at inference time is aware of the poison digits, so they can ask “is this string of 12 digits the poison digits?” and if not, it is likely to be partially poison digits. At a very high level, your question is touching upon the larger research question of how to tell when LLMs are behaving in their retrieval mode, by generating the secret digits they have already seen, and when they are behaving in their hallucination mode, by generating a string of 12 digits they have never seen. Therefore we can expect that advances in the larger research area can also be applied here, because if we can enforce that the LLM never hallucinates, that is, it only ever outputs a string of 12 digits that was present in the training data, and if we assume that the only strings of 12 digits are the poison and secret digits, then the attacker can always know whether the LLM’s generation is the correct secret digits or not, because if not then it will always be the poison digits, as any other generation would contradict the enforcement of no hallucination.
>
> > Figure 8: I think...
>
> **Question 4:** We have added more discussion in Appendix C, the paragraph titled ``Expanding ablations in main paper'' and Figure 15. We believe this is related to the above answers regarding overfitting to poisons. The bump is because allowing the strength of the poison to degrade reduces the overfitting to the poison digits. We don’t make a big point of this because as we mention, the attacker has no control over this parameter. To clarify, the fact that the relationship between, a) the number of iterations that pass between inserting the poison and observing the secret, and b) the SER, is convex is interesting and may require future investigations, but there are entire papers written about analyzing this concept of ‘durability’ and we are already a bit limited on space.
>
> > Figure 8: Is all the poison inserted all together?
>
> **Question 5:** For computational efficiency we insert the poisons altogether. If we were to instead spread all the poisons randomly throughout the train loader, we would have to train for many more iterations (>10x) or insert more poisons. One reason why we are able to perform so many ablations is because we can just insert all 100 poisons into 100 consecutive iterations in the train loader.

---

> > ### Comment · Reviewer_bBB6 · 2023-11-22
> >
> > > Question 1
> >
> > I see. My original intent was to understand why "not" and not any other string. If it is just the case of "we tried this first and it worked" that's fine, but should mention that.
> >
> > > Question 2
> >
> > Great! Glad to see it helped.
> >
> > > Question 3
> >
> > This is a fair point, and indeed touches upon hallucination and other relevant concepts in LLMs.
> >
> > Overall, I think the paper has improved greatly, both in terms of included experiments and results, as well as clarity in assumptions. Considering this, and the fact that my main concerns around the paper have been mostly addressed, I am increasing my score.

---

> > > ### Author Response · Authors · 2023-11-22
> > > **Response to Reviewer**
> > >
> > > Thank you for your detailed response, increasing your score, and thank you again for the thoughtful review. We really appreciate the time you took to review the paper and help us improve it.
> > >
> > > We will just address the remaining weaknesses/comments/questions altogether.
> > >
> > > Comments:
> > >
> > > > I think this is interesting and would encourage the authors to at least mention this in their text.
> > >
> > > Thank you, we will add this to the text.
> > >
> > > > Indeed, this was my intent when bringing up models tuned no human preferences data. Adding a sentence behind the choice for Pythia (including a record of exact data and when it was seen) would be helpful.
> > >
> > > Thank you, we will add this to the text.
> > >
> > > > Yes, I know what the formula is - I mean to ask how you compute this similarity between tokens. For instance, how would you do it for "I am a dog" and "The sky is blue"? Do you just convert tokens to one-hot vectors and do it, or consider sentence-level embeddings?
> > >
> > > We apologize for not providing sufficient detail. The full code for computing the cosine similarity can be found in Appendix F, so we did not provide details on the pseudocode. We compute the sentence-level embedding of the entire prefix, using the openAI text-ada-002 embedding model because it is publicly available and commonly used.
> > >
> > > Weaknesses:
> > >
> > > > While I understand the 'computational efficiency' limitation, I personally would've like to see cases where the adversary had to wait for even more iterations, or where poison insertion was scattered across training runs. At the same time, I also want to keep in mind that not all researchers (including myself) would have access to sufficient compute to run such experiments :)
> > >
> > > Thank you for being understanding. We think this can be something to study in follow-up work with different datasets. As per Figure 10, we test waiting for up to 10k iterations, because after 10k iterations the adversary can't extract any secrets, so we don't test for longer waiting times. This may be because The Pile datasets are pretty easy for this model to learn, and of course the actual fine-tuning datasets that people use (like OpenHermes) are much more complex and/or larger.
> > >
> > > Questions:
> > >
> > > > I see. My original intent was to understand why "not" and not any other string. If it is just the case of "we tried this first and it worked" that's fine, but should mention that.
> > >
> > > Thank you; we will add this (it was the first thing that we tried to fix the problem, and it worked, so we just went with it).
> > >
> > >
> > > All the changes described above should be in the text by the AoE deadline today. We thank you once again for the review and for recommending an acceptance for our paper.

---

### Author Response · Authors · 2023-11-21
**General Response to Reviewers**

We greatly appreciate the time and care that the reviewers have put into the reviews. We thank you all for your constructive feedback, which we have incorporated into the new version of the paper. This includes new versions of all figures, updates to the text, and a new Appendix B (Defenses) and Appendix C (Multi Secret Extraction).

---

### Meta-Review · Area_Chair_WRfw · 2023-12-08

**Metareview:**

This paper proposes a method for amplifying data extraction attacks against LLMs. The authors show that an adversary can dramatically increase the extraction rate by injecting poisoned data into the model's pre-training set. Upon fine-tuning the model on private data, the model is more likely to regurgitate memorized private information. Through careful ablation study, the authors also analyze the effect of model size, data duplication, etc. on their finding.

Reviewer Lw6r raised concerns about the practicality of the attack's setting, requiring that the adversary knows the exact secret prefix and can prompt the model shortly after fine-tuning on the secret, and that the attack only works on pre-training data. There is also prior work [1] that observed similar findings. In the end, the reviewer did not consider their concerns to be crucial weaknesses, and the AC recommends acceptance but strongly encourages the authors to discuss similarities and differences with the referenced prior work [1].

[1]: Controlling the Extraction of Memorized Data from Large Language Models via Prompt-Tuning. Mustafa Safa Ozdayi, Charith Peris, Jack FitzGerald, Christophe Dupuy, Jimit Majmudar, Haidar Khan, Rahil Parikh, Rahul Gupta. https://arxiv.org/abs/2305.11759

**Justification For Why Not Higher Score:**

While the paper's findings are interesting and significant, they are somewhat expected and prior work suggested by Reviewer Lw6r observed similar findings.

**Justification For Why Not Lower Score:**

This paper presents interesting findings that highlight a realistic threat, with careful empirical evaluation to back up its claims. The only negative review is only slightly negative, and does not object to the paper's acceptance.

---

### Decision · Program_Chairs · 2024-01-16

Accept (poster)